# Cause, Regulation and Utilization of Dye Aggregation in Dye-Sensitized Solar Cells

**DOI:** 10.3390/molecules25194478

**Published:** 2020-09-29

**Authors:** Fang Xu, Thomas T. Testoff, Lichang Wang, Xueqin Zhou

**Affiliations:** 1School of Chemical Engineering and Technology, Tianjin University, Tianjin 300345, China; xufang31@126.com (F.X.); lwang@chem.siu.edu (L.W.); 2Department of Chemistry and Biochemistry and the Materials Technology Center, Southern Illinois University, Carbondale, IL 62901, USA; thomas.testoff@siu.edu

**Keywords:** dye-sensitized solar cell, dye aggregation, DFT methodologies

## Abstract

As an important member of third generation solar cell, dye-sensitized solar cells (DSSCs) have the advantages of being low cost, having an easy fabrication process, utilizing rich raw materials and a high-power conversion efficiency (PCE), prompting nearly three decades as a research hotspot. Recently, increasing the photoelectric conversion efficiency of DSSCs has proven troublesome. Sensitizers, as the most important part, are no longer limited to molecular engineering, and the regulation of dye aggregation has become a widely held concern, especially in liquid DSSCs. This review first presents the operational mechanism of liquid and solid-state dye-sensitized solar cells, including the influencing factors of various parameters on device efficiency. Secondly, the mechanism of dye aggregation was explained by molecular exciton theory, and the influence of various factors on dye aggregation was summarized. We focused on a review of several methods for regulating dye aggregation in liquid and solid-state dye-sensitized solar cells, and the advantages and disadvantages of these methods were analyzed. In addition, the important application of quantum computational chemistry in the study of dye aggregation was introduced. Finally, an outlook was proposed that utilizing the advantages of dye aggregation by combining molecular engineering with dye aggregation regulation is a research direction to improve the performance of liquid DSSCs in the future. For solid-state dye-sensitized solar cells (ssDSSCs), the effects of solid electrolytes also need to be taken into account.

## 1. Introduction

The properties of a material not only depend on the physical and chemical properties of the elementary molecules that comprise it, but also on the aggregation patterns of molecules to a large extent. Molecules tend to aggregate and assemble into ordered structures through the complex synergistic effect of intermolecular interaction, thus enabling aggregates that have new functions [1]. At present, the research on molecular aggregation has covered almost all the frontiers of chemistry, as well as other fields such as biology, medicine, materialogy and mathematics. Especially in the field of photoelectric materials, with the discovery of aggregation-induced emission phenomenon [2], the research on organic molecular aggregation has once again set off a great upsurge.

As a core member of the field of optoelectronics, the solar cell is an important way to solve the energy and environmental crisis by using green energy, especially organic solar cells (OSCs), which have variety materials, low energy consumption, and can be printed in large areas at low cost [3,4]. As the third generation of OSCs, DSSCs use low-cost metal oxide and photosensitive dyes as the main raw materials to convert solar energy into electricity by simulating the photosynthesis of plants using solar energy in nature. It is apparent that the superiority of DSSCs lies in the low-cost and availability of raw materials, long life of the device, and large-area preparation. Owing to factors the aforementioned DSSCs’ have a tremendous potential for commercial applications [5]. DSSCs can theoretically be divided into two categories, n-type DSSCs and p-type DSSCs; n-type DSSCs research accounts for the vast majority due to its superior battery efficiency. The obtained power conversion efficiency (PCE) of n-DSSCs has grown from 7% [6] to 13% [7] using I_3_^−^/I^−^ redox couple additive electrolyte, and the highest PCE of 14.3% was achieved in cobalt-mediated DSSCs [8]. In accordance to their importance to the structure of DSSCs, most of the study is mainly focused on nano-porous semiconductor thin films, counter electrode and sensitizers [9,10], specifically the research pertaining to molecular aggregation. The research on the nano-porous semiconductor thin films is diversiform and well-studied; materials with different morphologies, such as nanoparticles, nanorods, and nanotubes, have been developed by regulating molecular aggregation patterns to improve device performance [11,12,13,14,15]. For DSSCs, platinum counter electrodes give the best performance, but their high cost restricts the commercialization. Therefore, researchers pay more attention to the development of low cost and high activity nano carbon materials, conductive polymers and their composite materials, etc. [16,17,18]. It is worth noting that in recent years, flexible DSSCs assembled on non-planar fibrous conductive substrates with high curvature and excellent stitchability have been constantly developed, which break through the limitations of traditional flat substrates and have the advantages of bendability, light weight and wide application prospects [19,20,21,22]. As the most important core part of DSSCs, the development of new dyes has always been the mainstream focus of DSSCs research, and people have gradually turned their attention to the study of dye aggregation in recent years.

Dye aggregation is of paramount importance in determining the overall DSSC conversion efficiency [23], short-circuit current (J_sc_) [24,25], open-circuit voltage (V_oc_) [26,27] and fill factor (FF) [28] of each device. In general, aggregation occurs during sensitization and can be classified as H-aggregation and J-aggregation. Dye aggregation may lead to quenching of the excited state, so many methods such as molecular engineering, the use of co-adsorbents, the alteration of sensitization conditions, etc., have been used to inhibit the aggregation [29]. Meanwhile, the advantages of dye aggregation cannot be ignored, especially J-aggregation, which is conducive to light absorption in the near-infrared spectrum, and it will be a potential pathway to improve the efficiency of DSSCs, though there are few reports at present [30,31,32]. Some materials characterization methods were used to study the aggregation patterns of dyes, including UV/vis and fluorescence spectroscopy [33], scanning tunneling microscopy (STM) [34,35], atomic force microscope (AFM) [36], X-ray reflectometry (XRR) [37], etc. And with the development of quantum chemistry theory and computer technology, we can better elucidate the structure and properties of dye aggregates at the molecular level [38].

The objective of this review is to provide a broad overview of the current studies of dye aggregation in DSSCs. Herein, we unfold the operational principles and structure of DSSCs, discuss the formation and influencing factors of dye aggregation, and the current research trends are reviewed. Finally, the application of quantum computation in the study of dye aggregation is described.

## 2. Operating Principle and Characterization Parameters of DSSCs

### 2.1. Operating Principle

The device structures of n-DSSCs and working steps are illustrated in Figure 1a. Traditional n-type DSSC devices are mainly composed of a transparent conductive glass substrate, a porous metal oxide semiconductor film (commonly, TiO_2_, ZnO, SnO) adsorbed with photosensitive dyes, electrolyte containing a redox couple (typically, Iˉ/I_3_ˉ) and counter electrolyte [39,40]. Solid-state dye-sensitized solar cells (ssDSSCs) have a solid hole transport material (HTM) in place of the electrolyte [41,42]. The fundamental processes are introduced as follows:The ground-state dye molecules adsorbed on the metal oxide surface are excited by light, and the electrons transition from highest occupied molecular orbital (HOMO) to lowest unoccupied molecular orbital (LUMO).Excited electrons are injected into the conduction band of the metal oxide, then electrons migrate to the conductive substrate, and enter the external circuit to form a current.Regeneration of the oxidized dye by electron donation from the redox couple of the electrolyte.The oxidized species in the electrolyte receive electrons from the external circuit to complete the process.

For n-ssDSSCs, the dye regeneration is attributed to direct hole transfer from the oxidized dye molecule into the HTM, whereas the redox reaction in a liquid-state system is diffusion-limited. The HTM is then regenerated by charge transfer at the counter electrode [43].

The structure of p-type DSSC devices and working steps are illustrated in Figure 1b. Its charge transfer process is opposite to n-type DSSC [44]. The most frequently used p-type semiconductor is NiO, while other inorganic materials also have been investigated, such as CuO, etc. [45,46,47].

### 2.2. Characterization Parameters of Cell Efficiency

The overall power conversion efficiency (PCE) are evaluated by J_sc_, V_oc_, FF and the incident light power (P_in_).
(1)$$\eta =\frac{J_{\mathrm{sc}}V_{\mathrm{oc}}FF}{P_{\mathrm{in}}}$$

J_sc_ is the current density measured by without applied external bias. It is determined not only by the molecular structure of dyes, but also by the amount of adsorption onto porous metal oxide semiconductor films as well as the electrochemical properties of the porous film [48,49]. V_oc_ is determined by the potential difference between the quasi-Fermi level of the semiconductor and the redox level of the electrolyte. It is influenced by the recombination rate and adsorption mode of the dye [50]. FF is defined by the ratio of the maximum power of the solar cell to the product of J_sc_ and V_oc_ and is introduced to account for non-ideality of the I-V curve. This parameter is influenced by the electrode materials, active materials, charge transfer resistance between interfaces and battery package [51,52]. Moreover, the incident-photo-to-current conversion efficiency (IPCE) is usually utilized to evaluate the spectral response of solar cells and defined as the ratio between the photocurrent density produced in the external circuit under monochromatic illumination of the cell and the incident photon flux.
IPCE = 1240 J_sc_/(λ P_in_)(2)
where λ (nm) and P_in_ (mW cm^−2^) are the wavelength and intensity of incident monochromatic light. Considering the current generation process, IPCE is jointly determined by the light harvesting efficiency (LHE), the electron injection efficiency φinj and the charge collection efficiency φcoll at the electrode [53,54]. The formula is as follows:IPCE (λ) = LHE(λ) φinj φcoll(3)

In general, the overall conversion efficiency of DSSCs is tested under standard irradiation conditions (100 mW cm^2^, AM 1.5). Over the past decade, we have investigated various dye molecules and their derivatives in the attempt to understand and tune their photophysical properties to maximize the cell efficiency [55,56,57,58,59,60,61]. These studies and the experience in fabricating solar cell devices have demonstrated that any significant improvements in cell efficiency of DSSCs cannot be achieved without in-depth studies of dye aggregation.

## 3. Dye Aggregation in DSSCs

### 3.1. The Mechanism of Dye Aggregation

It is well-known that the molecules in real compounds do not exist as single molecules, some of them form dimers, trimers and other higher order aggregates. Unlike crystalline solids, which have long range orderliness, organic materials that aggregate mainly depend on different intermolecular forces. Generally, the aggregation between dye molecules can be classified as H-aggregates (face-to-face arrangements) and J-aggregates (edge-to-edge arrangements) [62], which can be explained by Kasha’s molecular exciton theory [63]. Based on excited state resonance interaction, molecular exciton theory describes a resonance splitting of the excited state composite molecule energy levels and expound the relationship between the spectral properties of molecular aggregates and molecular structure (Figure 2). When two molecules interacting to form a dimer, coupling phenomena lead to excited-state energy-level splitting. For the face-to-face alignment of molecules, the molecular transition dipoles corresponding to the higher excited state E″ are arranged in the same direction, and the oscillator strength of the radiation transition is concentrated in the exciton state E″, thus, the hypsochromic shift occurs in the absorption spectrum. The transitions from the ground state to exciton state E′ are forbidden, while transitions from the ground state to exciton state E″ are allowed. In contrast, for the edge-to-edge alignment of molecules, the molecular transition dipoles corresponding to the lower excited state E′ are arranged in the same direction, so the transition from the ground state to the exciton state E′ are allowed, and the bathochromic shift is observed in the absorption spectrum. In addition, the type of aggregates can be judged from the angle(α) between the transition dipole moment and the line of molecule centers, i.e., H-aggregate (54.7° < α < 90°) and J-aggregate (0° < α < 54.7°).

### 3.2. The Influence Factors of Dye Aggregation

The main cause of molecular aggregation is intermolecular interaction, which is generally weak, such as van der Waals force, hydrogen bond, aromatic ring stacking, hydrophobic interaction, etc. When the aggregate is formed, it affects the internal movement of each molecule in the aggregate, resulting in new characteristics that single molecules do not have [64,65]. The formation of aggregates is the manifestation of the overall effect of the complex synergy of various weak interaction forces. Therefore, the aggregation of molecules is not only related to chemical structural characteristics (decide on the possibility of achieving some kind of aggregation), but also changes with the environment in which the molecules are located (influence the formation of aggregated structures).

Aggregation of dyes is strongly affected by dye structure, concentration of dye bath, ionic strength, temperature, solvents, and so on [66,67]. It occurs in the sensitization process, mainly including the following two processes: (1) The dye first aggregates in solution (caused by the interaction among dyes and/or other molecules) and then adsorb on the metal oxide semiconductor; (2) The dye aggregates after the dye has been adsorbed on the metal oxide semiconductor surface to form the dye/metal oxide semiconductor interface. However, dye aggregation usually causes the quenching of molecular excited states and unfavorable back electron transfer, which seriously affects the electron injection efficiency, and then diminishes the device efficiency [68,69,70]. Besides, H aggregation will cause a blue-shift of absorption spectrum, which is not conducive to improving the light absorption capacity. Thus, most studies focus on improving the performance of the device by inhibiting aggregation. At present, the methods for inhibiting dye aggregation mainly include changing the structure of dye molecule, regulating the conditions of the dye sensitization process, and changing the dye/metal oxide semiconductor interface environment.

### 3.3. Methods for Inhibiting Dye Aggregation

#### 3.3.1. Molecular Engineering

Molecules with good planarity are prone to H aggregation, so reducing the planarity of dyes can effectively inhibit aggregation. A twisted structure of π-conjugated organic dye **1** (shown in Figure 3) was synthesized to inhibit aggregation, resulting in a 67% increase in device efficiency (see Table 1) compared to the corresponding planar dye **2** [71].

Currently, the introduction of flexible chains and bulky groups are the most commonly used methods to suppress dye aggregation in DSSCs. The presence of long flexible chains can increase the distance between molecules on TiO_2_ surface or change the planarity of molecules [72], which results in reduced dye aggregation. Aggregation of dyes on the TiO_2_ surface depends on the length and the location of the flexible chains. A series of unsymmetrical squaraine dyes (**3**–**8**) with different alkyl chain lengths at the indoline unit were synthesized to modulate the aggregation of sensitizers on the TiO_2_ surface, which resulted in an overall PCE ranging from 3.82% to 6.23% (see Table 1) [73]. In addition, the long chain not only reduces the aggregation, but also prevents the electrolyte from penetrating the dye [74].

Quinoxaline-based sensitizers **9** and **10** with long alkyl chains on their auxiliary electron acceptors successfully suppress dye aggregation and electrons recombination. Meanwhile, the incorporation of alkyl chains into donor group broadens the light-harvesting in the long-wavelength region [75]. Zhu and co-workers suggested that the hexyl chains on the bridged thiophene rings help to avoid dye aggregation on the NiO film and block I^−^ in electrolyte from approaching the surface of NiO [106].

Tuning the length and number of alkyl/alkoxy chains, a substantial enhancement of PCE from 5.2% to 9.1% (see Table 1) was achieved in a series of donor-π-acceptor-type porphyrin dyes (**11**, **12** and **13**), due to the aggregation being inhibited by introducing flexible chains into suitable positions without seriously aggravating distortion of the dyes [76]. The photovoltaic performance comparison of ssDSSCs using triphenylamine-based dyes 14 and 15 as sensitizers demonstrate the beneficial influence of alkoxy units, enhancing light harvesting and resulting in a higher photocurrent. Various intensities of light were used to test photovoltaic performance of ssDSSCs, and under low intensity (10% sun and 50% sun), the lower observed photocurrent is attributed to aggregation of dyes on the TiO_2_ surface [101]. A dye packing model is proposed to reveal the impact of dye aggregation on the overall photovoltaic performance, which has also been reported in p-type DSSCs [107].

Porphyrin sensitizers have a relatively strong tendency toward aggregation induced by their large conjugated framework [108,109]. Sensitizers **16**, **17,** and **18** were designed by introducing bulky dihexyloxyphenyl groups to suppress dye aggregation and improve the photovoltage, affording a highest efficiency of 7.03% (see Table 1) [78]. With the six hexylsulfanyls as the bulky electron-donating groups, sensitizer **19** suppresses dye aggregation on TiO_2_, and the higher electron donating ability was favorable for light harvesting [79]. But at the same time, the introduction of bulky groups will also decrease the loading amounts. A methyl group was introduced into the acceptor, effectively suppressing dye aggregation, and thus simultaneously improving the J_sc_ and V_oc_ values. Meanwhile, the introduction of the methyl group had no obvious unfavorable steric hindrance and slightly increased the dye-loading amounts [110].

Another molecular engineering technique to control the aggregation of dye on TiO_2_ was realized by anchoring dyes on the surface with suitable anchoring groups [111]. However, the effect of anchoring groups on dye aggregation haven’t been extensively investigated, and most of them focus on theoretical research. The anchoring group mainly effects the molecular space stretching (Figure 4) and anchoring mode (adsorption site [112] and orientation [113], etc), and the intermolecular interaction between the anchoring groups have a direct influence aggregation [114,115].

Position engineering of anchoring group in a dye demonstrates its influence on the molecular space stretching, and the different dye aggregation causes differences in device FF [116]. The number of carboxylic acid anchoring groups in the dye is correlated to the extent of dye aggregation [117]. Due to carboxylate ions that are formed during the dye sensitization process and carboxylic acids will interact strongly with other chemical substituents to promote molecular aggregation [118]. Orduna et al. reported three dyes with different anchoring groups (**20**, **21** and **22**) (shown in Figure 5) for ssDSSCs. From the graph of dye bath absorption intensity and sensitization time, it can be known that the concentration of dye **20** kept decreasing over the entire time. This trend may arise from undesired dye aggregation at the TiO_2_ surface. Notably, the concentration of dye **22** in solution decreased rapidly and then saturated, i.e., does not show any evidence of this type of dye aggregation, which may be a partial cause for relatively high J_sc_ (see Table 1) [102].

Apart from the above methods, dyes containing two anchoring groups were synthesized to avoid aggregation utilizing conjugate/configuration linkers through either donor or π-spacer component of the sensitizer (Figure 6). Compared with the corresponding single-branch dyes, the double branched dyes possess better broad absorption, stronger binding ability, and reduced tendency towards aggregation [119,120].

The photophysical and photovoltaic properties are influenced by the distance between two anchoring units and flexibility of the linker. For example, dyes **23** and **24** (shown in Figure 7) linked two donors using m-phenylene and fluorene units, the trapezoidal **24** exhibited lower aggregation and charge recombination than the inter-planar **23** and achieved a broader light-harvesting range [80]. A series of phenothiazine-based dyes (**25**, **26** and **27**) have lower tendency to aggregate due to the characteristic of the double branch, and the position of the linkage units influences the photophysical characteristics, dye loading on films, and the electron lifetime [81]. Double branched triphenylamine-based dyes (**28**, **29,** and **30**) with bridge linked at different position of the π-bridge also have been designed, and the cross structure is superior in the suppression of intermolecular interactions, reducing the charge recombination rates in the DSSCs [82]. The PCE of DSSCs has been greatly improved compared to the corresponding isomers. A meso-meso directly linked porphyrin dimer (dye **31**) has been shown to reduce intermolecular π-π interaction and suppress the approach of I_3_^−^ electrolyte ions to the TiO_2_ semiconductor for charge recombination [83]. With respect to the corresponding non-spiro-linked parent compounds, spiro compounds have higher solubility, improved morphological stability in the solid-state and a reduced tendency to form aggregates [121,122]. The homodimeric spiro dye 32 showed higher PCE of 7.6% (see Table 1), which is 2.4 times the monomeric form, and the spiro linker offers the dye better flexibility [84].

Iyer and co-workers reported a A-type *o,m*-di fluoro substituted phenylene spacer di-anchoring dye **33**, which exhibited a strong ICT peak as both acceptor units help in efficient electron extraction from the carbazole donor and reduced H-aggregation in solution as well as in solid-state. Compared with the mono-anchoring dye, dye **33** has a higher PCE in the presence of iodide redox electrolyte, but exhibits comparatively less efficiency when used to fabricate a ssDSSC due to aggregation between dye and solid organic ionic conductor through H-bonding [85]. Lin et al. reported a new series of A-type phenothiazine-based dye with double acceptors/anchors (**34**, **35** and **36**), i.e., two branches share one donor group. Compared to the congener containing only one anchor, they not only more efficiently suppressed dark current and dye aggregation, but also provided more electron injection pathways [86]. The substituents at the nitrogen atom of phenothiazine further contributed to the inhibition of aggregation [86,123,124]. For A-type symmetric dyes, the functionalization of a central core can not only force the absorption geometry of the dyes [125], but also greatly change the spatial geometry of molecules and geometry of dyes chemisorption on semiconductor surface, thus affecting the dye aggregation on the semiconductor surface [126].

#### 3.3.2. Co-Adsorbents

Co-sensitization by adding co-adsorbing agents to the dye bath during the sensitization process is also a broad class of methods commonly used to inhibit dye aggregation. Cholic acid (CA), deoxycholic acid (DCA), chenodeoxycholic acid (CDCA) and their derivatives are commonly used as co-adsorbents [127,128,129]. The co-adsorbents, with fewer restrictions and frequent use, can not only effectively inhibit the formation of the undesired aggregates to reduce excited state quenching [130,131,132], but also cover the exposed TiO_2_ surface between dyes, which could protect the dye-TiO_2_ working electrode from electrolyte attack, and then reduce charge recombination [133,134]. Highly selective co-adsorbents are also designed to improve the performance of DSSCs through effective spectral complementation and suppression of dye aggregation [135,136,137], which are often seen in porphyrin dyes.

In previous studies, CDCA and its derivatives are frequently used as co-adsorbents to enhance the efficiency of DSSCs by inhibiting dye aggregation (Figure 8). In the absence of co-adsorbents the aggregation adversely impacts the performance of DSSC, but, since the co-adsorbent is a competitor to the dye adsorption, a large concentration decreases the dye loading and consequently J_sc_ [138,139,140]. So, there is an optimal concentration of co-adsorbent in the solvent bath. In recent years, research on CDCA and its derivatives is no longer aimed at improving device efficiency, but is more used to study the types, regulation, and verification of dye aggregation.

CDCA and HC-A1 were co-adsorbed with Zn-porphyrin dyes **37**, **38**, and **39** (shown in Figure 9) respectively, and the addition of both co-adsorbents improves the PCEs of all sensitizers. The DSSCs with HC-A1 exhibited higher J_sc_ and PCE (see Table 1) compared with CDCA due to the reduced J-aggregation and improved light-harvesting ability in the 350–480 nm region [87]. 

The steroid backbone of bile acids was harnessed to chemically modified co-adsorbents CA, DCA and CDCA at their R3 position attaching sterically demanding amide groups. All co-adsorbents regardless of the concentration ratio furnish better efficiencies compared to devices without them, which correlates with a valuable effect of decreasing dye aggregation [141].

Zhu et al. adopted the strategy of co-sensitization to improve the device efficiency of benzothiadiazole-based dye **40**. Co-adsorption with CDCA reduces dye aggregation, and the amount of adsorbed dye **40** has almost no significant change, but there is no dramatic promotion in photovoltaic efficiency. For the co-sensitized cells of **40** and **41**, the IPCE spectra showed above 80% from 450 to 650 nm with the cutoff wavelength extending to 800 nm, achieving an excellent efficiency of 9.83% (see Table 1) [88].

Two organic co-sensitizers S3 and S4 containing bipyridine anchoring groups were used with porphyrin dye **42** for DSSCs. The co-sensitizers not only suppress the dye aggregation and fill the adsorption gap between the main dyes, but also successfully compensates for the absorption defects of porphyrin dyes and improves the light response current [142]. As reported by Harrabi and co-workers, it should be noted, the performance of the co-sensitization DSSCs depends strongly on the concentration of the co-sensitizer [143].

And the same approach is also used in ssDSSCs [144,145]. Grätzel and co-workers reported a near-IR squaraine-based dye **43** for ssDSSCs. By effectively reducing the dye aggregation via adding CDCA in the dye bath, the aggregation peak is decreased, and the absorption peaks are slightly red-shifted. Interestingly, the addition of spiro-MeOTAD has a similar effect to CDCA [146]. Buddhapriya and co-workers showed that replacement of the ethyl group on the terminal rhodamine ring of indoline dye **44** by octyl substitution does not completely stop aggregation. The dye aggregation was further inhibited by adding co-sensitizers, and maximum PCE of 3.6% (see Table 1) was observed for TiO_2_/dye/CuI ssDSSCs [103]. For ssDSSCs, the solid hole transport layer itself makes a certain contribution to the inhibition of dye aggregation. The process of depositing the HTM by spin-coating may remove the aggregates not firmly attached to the TiO_2_ surface or disrupt the dye aggregates [146].

It is well-known that CDCA is generally used to inhibit dye aggregation in DSSCs, as many of the studies mentioned above, but it is very interesting that Kothandaraman and co-workers reported two pyrene carbazole dyes **45** and **46**, which showed opposite effects after co-adsorption with CDCA. On TiO_2_ film, **46** dye, along with CDCA, show a slight broadening in their absorption spectrum compared to CDCA free solid-state spectrum, whereas for **45**+CDCA, a significant blue-shift was observed. Also, **46** showed an enhancement of PCE upon the addition of CDCA, while **45** showed a decrease, which was directly related to the change of FF (see Table 1) [91].

#### 3.3.3. Sensitization Conditions

Sensitization is an important step in the preparation of DSSCs; dye baths (solvents and their concentrations) have an important impact on dye aggregation by affecting the anchoring mode, adsorption capacity and absorption spectrum of dyes [147,148,149]. By influencing the anchoring mode of the dye, the solvent can control the arrangement mode on the TiO_2_ surface to achieve the purpose of regulating aggregation. Compared with the above two methods of molecular engineering and co-sensitization, it is more convenient to regulate the aggregation of dyes through solvent selection. Rensmo et al. reported that water in the dye solution greatly reduced ZnO surface dye aggregation and thereby enhances the solar cell performance for N719 by using XPS (see Table 1) [90].

The dyes in different solvents exhibit diversified interactions between the dyes and solvents [150], which could cause changes to the physical and chemical properties between the dyes and semiconductor surface. Triphenylamine-based dye **47** (shown in Figure 10) was employed to sensitize TiO_2_ in different solutions (CH_2_Cl_2_, ACN, EtOH, THF, and DMF) [92]. The decrease of photocurrent density of DSSCs via different solutions is in direct proportion to the decreasing adsorbed amount.

Two solvents (CH_2_Cl_2_ and CHCl_3_) were used to study the dye-bath solvent effect on aggregation effect of benzothiadiazole-based dye **41** (shown in Figure 9). The augmentation of surface coverage in CH_2_Cl_2_ does not lead to an increase of the photocurrent. Non-polar solvent CHCl_3_ appears to solvate the dye better, resulting in lower dye loading, but in better spatial distribution, preventing the undesired π-π stacking, which increased PCEs from 3.86% (CH_2_Cl_2_ solvent bath) to 7.22% (see Table 1) [89].

It is well-known that some organic dyes can exhibit positive solvatochromism that causes a bathochromic shift of the absorption band with increasing solvent polarity, and some dyes show negative solvatochromism that can lead to a hypsochromic shift [151,152,153]. Therefore, the polarity of solvents has a great influence on the aggregation pattern of dyes. By dissolving more than 3000 dyes in different solvents, the fingerprint-based classification models proposed by Venkatraman is demonstrated in experiment. Also, in the collected data, highly polar solvents such as ethanol, methanol, and dimethylformamide show a higher incidence of J aggregation, while those on the mid-lower range of the scale, such as chloroform and tetrahydrofuran, show mixed behavior [154].

The interaction between dye molecules and solvents also has an important effect on dye aggregation. For alkyl-functionalized carbazole dye **48**, a clear distinction in the absorption peaks in two cases indicate a difference in the aggregative blue-shift in toluene due to J-aggregation and the red-shift in ACN due to H-aggregation. The higher affinity of the hydrophobic moiety due to toluene account for the difference [105].

In addition to the dye bath itself, the sensitization time and temperature also affect the formation of dye aggregation [145,155,156]. The length of time determines the amount of dye adsorption, and aggregation prevails with time at the expense of the monomer species due to the increasingly confined surface area [157]. For example, Park et al. prolonged the adsorption times (quinoxaline-based dye **49**), and it showed reduced efficiencies and Jsc values (see Table 1), resulting from dye aggregation and intermolecular quenching [93]. Upon acidification of the solution [158] and change of anions in the electrolyte [94], the H-aggregation of porphyrin-based dye **50** molecules is converted to J-aggregation with higher photo-to-electron conversion efficiency.

#### 3.3.4. Other Methods

Different from traditional co-sensitization, in which the co-adsorbents are all anchored to the semiconductor surface via functional groups, Miyasaka and co-workers presented a concept of co-sensitization of dye **44** (D149) with methylammonium lead bromide (MAPbBr_3_) perovskite in ssDSSCs. Small amount of MAPbBr_3_ can mitigate dye aggregation and can improve the light harvesting property of device. Also, the dye/perovskite co-sensitized device showed efficient charge transfer between dye and TiO_2_ (Figure 11) [104].

In addition to the above three methods (molecular engineering, the use of co-adsorbents, and the change of sensitization conditions), some special methods have also been used to inhibit dye aggregation, one of which is to add additives in the electrolyte. Chemical modification of a dye **51**-adsorbed TiO_2_ photoelectrode is achieved by adding MPA in the electrolyte to replace the protons in the free anchoring group of dye **51**, forming an amide bond (Figure 12). This method significantly retarded dye aggregation and resulted in enhanced J_sc_ (see Table 1) [95].

Similar to the above ideas of dye modification, another dye-modification material is proposed. However, the latter does not directly change the dye structure through covalent bond. After sensitizing with dyes, the TiO_2_ electrode was dipped into a solution of Al(OC_3_H_7_)_3_, then hydrolyzed to Al_2_O_3_. Multi-layer DSSCs can be obtained by repeating the above steps. Also, this alternating assembly structure can increase the adsorption of dye sensitizer **51**, while prohibiting the dye aggregation at the same time [159]. Such atomic layer deposition also has been reported by Son et al. to minimize dye interactions caused by aggregation (triphenylamine-based dye **14**). It is more effective than the widely used aggregation-inhibiting co-adsorbent CDCA and engenders a 30+% increase in overall energy conversion efficiency (see Table 1) [77]. Bian and co-workers also reported insulating Al_2_O_3_ layers, which contribute to bring about improvements in the performance of p-type DSSCs, the peak shift of absorption spectra and fluorescence spectra showed its effectiveness in inhibiting dye aggregation. The difference is that they first prepare the Al_2_O_3_/NiO films and then sensitize dyes [160]. In addition, the atomic deposition helps to passivate defects in NiO solar photocathodes [161].

Dye aggregation is caused by intermolecular forces, which depends on the distance between the molecules. In the above three methods of this section, when the dyes are anchored on the semiconductor metal oxide surface, the distance between molecules cannot be changed, and the post-treatment method is used by the researchers to suppress the dye aggregation, that is, adding a shielding layer between molecules. Changing the distance between the molecules to inhibit dye aggregation is a very effective and direct method, such as molecular engineering and the use of co-adsorbents. In addition, the electrode pretreatment has also been reported, which is a useful approach to reduce the number anchoring sites and, in turns, the dye aggregation. The electrode pretreatment with acids was reported by Wang and co-workers [162]. The strong acids such as HCl, HNO_3_ and H_2_SO_4_ increased the dye adsorption. However, HAc and H_3_PO_4_ decreased the dye amount on TiO_2_. It depends on the strength of anchoring groups, between the acid anchoring group and the dye anchoring group (-COO^−^). Tian et al. reported chemical and thermal methods to reduce the amount of Ni^3+^ in the NiO film. Also, the p-DSSCs built with the reduced NiO film showed an improvement in photovoltaic performance, especially in terms of photovoltage (see Table 1) [163].

The usual fabrication of DSSC devices is to prepare porous TiO_2_ thin films and then carry out dye adsorption. In terms of more dye adsorption and less dye aggregation, a new methodology of pretreated TiO_2_ with natural dye **52** was reported (Figure 13). Also, the modified colored TiO_2_ nanoparticles showed 47% improvement in efficiency (see Table 1) [96].

### 3.4. Future Prospects for Utilizing Dye Aggregation

It is well-known that the dye aggregation is of paramount importance in determining the overall PCE of DSSCs due to the optical and electronic properties that strongly depend on the aggregation patterns. Above all, the vast majority of studies focus on suppressing dye aggregation because the stronger intermolecular interactions within the compact dye layer leads to excited-state quenching, which will be detrimental to the electron injection efficiency. But the following points are worth noting: (1) The aggregation of dye molecules enables them to be arranged in order on the surface of TiO_2_, which facilitates efficient charge injection from aggregates [114,164] and causes higher absorption amount on TiO_2_ [165]; (2) Dye aggregation can effectively reduce the bare surface of TiO_2_, which will help with blockading the redox mediator from infiltrating into the TiO_2_ surface, and thus reduce electron recombination [166]; (3) Dye aggregation can broaden the absorption spectrum and enhance the ability of light-harvesting [167,168,169], especially J aggregation [64]. Although there is a competitive relationship between the advantages and disadvantages of dye aggregation, the prospect of using the advantages of dye aggregation to improve the overall performance of DSSC is considerable. At present, there are also a few relevant research reports.

The absorption spectra of the two dyes **53** and **54** as shown in Figure 14 containing thienothiophene and thiophene segments on TiO_2_ film are broadened, and 53 produces a slight blue-shift due to the formation of H-aggregate, **54** has a red-shift due to J-aggregation. Though **53** has a larger π-conjugation system than **54**, the **54** has a broader absorption spectrum due to J-aggregation, which is favorable for harvesting solar light and leading to a large photocurrent [97]. Venkateswararao et al. also reported that dye **55** (shown in Figure 14) containing carbazole as donor and π-linker has larger photocurrent, which was attributed to the longer wavelength light harvesting ability arising from the J-aggregation and the retardation of electron recombination with the electrolyte in comparison with dye **56** with bulky tert-butyl groups [98].

Lin et al. studies the effects of the presence of CDCA on the J-aggregated dye **57** (shown in Figure 14) and found that there was no change of efficiency. Compared with the dyes without aggregation, **57** has higher conversion efficiency when employed in DSSCs [99]. Hemavathi et al. also conducted similar experiments for D-A-π-A carbazole dye **58** (shown in Figure 14) and found that there was reduced efficiency after adding CDCA. The higher aggregation contributes to better absorption, leading to a higher current coupled with more population of CB states thereby having better V_oc_ and improved performance (see Table 1) [100].

According to absorption spectrum studies, different dye molecules have different concentrations of aggregation. The orderly arrangement of dyes on TiO_2_ surface is conducive to electron injection, and J-aggregation can increase the absorption of the long wavelength region [170]. 

For a series of squaraine dyes (**3**–**8**) (shown in Figure 3), although different alkyl chains were introduced to inhibit aggregation, the IPCE responses of the devices fabricated with six dyes were broad, which indicated that both the monomer and aggregated structures contributed to the charge injection process. Also, higher contributions to the IPCE response were observed from the aggregates rather than the monomeric form of the sensitizer [73].

Konno and co-workers showed that the photocurrent action spectrum of the TiO_2_/Indoline dye **44**/CuI is broad and commensurate with the adsorption spectrum of TiO_2_/Indoline dye **44** indicating that the J-aggregated dye effectively injects electrons to TiO_2_. An efficiency of 4.2% (see Table 1) indicating that organic dyes adsorbing strongly onto TiO_2_ and forming non-quenching aggregates are more suited for application in ssDSSCs [171]. The self-assembly of dye on TiO_2_ also can be used as a dense blocking layer between hole conductor and TiO_2_, which effectively avoids charge recombination [172,173].

## 4. Application of Quantum Computation in the Study of Dye Aggregation

Computational Chemistry is becoming an important tool to help understand the effects of dye aggregation on the absorption of solar energies and the subsequent electron transport in DSSCs. Here we focus on discussing the use of DFT calculations to obtain the properties, such as relative energy levels, and the quantities for describing charge transport processes in DSSCs [174]. Quantum Chemistry studies of DSSCs beyond DFTs can be found in the recent reviews [175,176,177].

### 4.1. Computational Methodologies

In the study of aggregation of dyes on a TiO_2_ surface, either a cluster model [178] or a slab model [179] is used to describe the TiO_2_ surface. The adsorption of dye molecules on the model TiO_2_ surface is then studied. DFT calculations of the adsorption of dye molecules to TiO_2_ or other oxides, such as Sn, Zn, Ni, and Cu oxides have often been carried out using DFT+U methods to accurately calculate absorption energies and band gaps [180,181,182]. For instance, using a (7 × 8) TiO_2_ cluster to represent the TiO_2_ surface, Feng et al. characterized multiple phenothiazine-based dyes (59, 60, 61, and 62 shown in Figure 15) containing 4 separate auxiliary chromophore donor groups on a TiO_2_ cluster. Utilizing the widely popularized B3LYP functional with the 6-31G(d,p) basis set to optimize the dyes, and subsequently using the MPW1K hybrid density functional with the same basis set for TD-DFT to generate a simulated absorption spectra. While their monomer calculations were relatively consistent with the experimental data in terms of absorption energy and electron injection energy (calculated with CAM-B3LYP), more importantly, utilizing DFTB optimized ground state absorbed structures, they employed MD simulations to probe the aggregation of the aforementioned dies on the TiO_2_ cluster surface. They were able to gauge the effect of donor units on the interaction energy of each species during aggregation. In addition, they calculated the intermolecular electronic coupling of the stacked dyes over the same period. They further used non-adiabitc marcus theory to calculate the reorganization energy of each species, as well as calculate the charge transfer rate between the stacked dimers [178].

Studies have shown that B3LYP is relatively inadequate at characterizing the interaction energy between stacked dimers unbonded to the surface, whereas ωB97xD has comparable results to CCSD(t)/CBS energies. Multiple methods of intermolecular interactions have been tried and are in relative agreement, including the “energy splitting in dimer” method as well as a direct approach. When studying the excitation energy of dimers, it has been found that the B3YLP-D method performs better than either ωB97xD or CAM-B3LYP, with the exception of systems where there is a prominent charge transfer character. Lastly, MPW1K is a viable option when trying to perform accurate TD-DFT calculations on dimeric systems [111,168,183,184,185]. The slab model is usually optimized by the commonly used PBE functional, whereas the optimization of the dyes themselves once absorbed to the surface is usually done with a dispersion-dependent functional, often a Grimme’s D2 or D3 correction [179,185,186,187,188,189].

The difference between B3LYP, and its Grimme’s D3 dispersion corrected analog were compared and it was shown that the interaction energy of B3LYP for two indoline base dyes **63** and **44** was nearly entirely repulsive (−0.1 kcal/mol and 1.42 kcal/mol for **63** and **44** respectively). However, with the application of the D3 dispersion correction, the interaction energy resembled the MP2 calculated interaction energy relatively well. The interaction energy for **63** and **44** for MP2 was −14.08 kcal/mol and −8.04 kcal/mol respectively whereas for B3LYP-D3 the interaction energy was −16.27 kcal/mol and −10.02 kcal/mol respectively [38]. Other studies have also corroborated the inefficiency of B3LYP in this regard [186]. It has also been displayed that it is possible to simulate the effect of electron injection of one dye onto the adjacent structure in the aggregate. By modeling the dye adjacent as a cation and coupling with TDDFT calculations a Stark’s shift consistent with the experimental data was seen in the absorption spectrum [38].

Multilayer aggregation models can also be readily tested as has already been completed with p-methyl red [178]. Here, not only is it shown that the relative changes in absorption spectra can be gauged due to differences in surface loading, but also changes to the adsorption energies between the dye and the surface due to increased intermolecular interaction with neighboring particles as well as a deeper understanding of the intermolecular interactions occurring within the aggregate. In the study of multilayer aggregation of p-methyl red the UV-vis absorption spectra was shifted 200 nm (something also seen in calculation of the monolayer) most likely due to deficiencies in the PBE functional as it neglects Hartree Fock exact exchange as well as severely underestimates the band gap.

Other studies have examined the loading on the surface and similarly examined the resultant changes in optical properties, binding energies, and the charge transfer properties due to different conformational arrangements and loading. Expectedly, as dye loading increases, so does the absorption intensity of the maximum absorption as well as a consistent red-shift that corresponds with increased loading of the TiO_2_ surface. However, upon introduction of a fifth species not bound to the surface, but rather held in place between the bound site by attractive dispersion forces, a blue-shift occurs and a decrease in absorption in relation to the tetramer bonded species. This blue-shift is consistent with the aforementioned shift seen in the multilayer model, which also has a second layer of dye loading held purely by weak forces. The decrease in peak intensity differs and is possibly due to the calculation accounting for dispersion-based interactions utilizing the Tkatchenko-Scheffler (TS) DFT-D scheme. Increasing the loading on the surface not only reduced the adsorption energy, but introduction of the weak force bound p-methyl red species between the four chemisorbed species also decreases the adsorption energy.

Aggregative structures have also been examied, using just the optimized monomer structure to do single point energy calculations varying the distance between each monomer on its z-axis to find the optimal distance between each monomer before rotating the molecule and sliding it along in either the x or y direction to find the most stable input configuration. Once this is done, the structure can be optimized at a lower computational cost. This has been done with various streptocyanines with the ωB97XD functional and the cc-pVDZ basis set, and the conductor polarizable continuum model (CPCM). Symmetry-adapted-cluster/configuration interaction (SAC-CI) calculations were performed to find the vertical excitation energies [190]. The accuracy of this type of calculation is highly dependent on the ability of optimizing functional used, to properly characterize the non-bonding forces between dimers. As such, the accuracy and relevance of this methodology is intimately tied to the development of accurate dispersion corrected functionals.

### 4.2. Functional Development to Accurately Describe Aggregations

The efficacy of DFT method in the study of dye aggregates [178,179,186,191] is reliant on the current computational methodologies’ ability to characterize the forces which differentiate aggregates from their molecular state, namely van der Waals (vdW) forces, but more specifically in the case of organic dyes, dispersion or hydrogen bonding [29,38]. Developing computational means to accurately characterize vdW interactions has been a pressing concern within the computational community, as most have found both ab initio and Density Functional Theory (DFT) methods lacking [192,193]. Through development of an accurate computational methodology it may be possible to calculate the changing absorption properties, in the case of sensitizers, of the species of interest upon aggregation [194,195]. This characterization may also be a boon to our understanding of the changing exciton mobilities in aggregated species as well, specifically modeling the changes in electronic structure that have been reported to cause electron mobility changes by three orders of magnitude in some cases [196,197].

In Kasha’s model, the alignment of the transition dipoles determines whether coupling is positive or negative. However, it has been noted by Spano that the Kasha model is purely limited to Coulomb coupling and, as such, does not factor vibronic coupling or wavefunction overlap. It also should be noted that not only is ground state geometry and energy affected by vdW interactions (lowering of ground state energy), but the subsequent vdW interactions between the excited and unexcited molecule also change the excitation energy in addition to the exciton splitting term. Equations (4) and (5) depict the ground and excited state energy respectively, where 1 and 2 refer to the individual monomers. The third term in the ground state energy equation is the vdW interaction energy and V_12_ is the intermolecular perturbation potential. The excited state energy is affected by an additional term called the exciton splitting term (E_split_) [63,198,199,200]. As such, in order to properly model and quantify dimer and aggregate transitions, it is necessary to properly quantify vdW interactions.
(4)$$E_{Ground}=E_{1}+E_{2}+\iint \;\varphi_{1}\varphi_{2}\left(V_{12}\right)\varphi_{1}\varphi_{2}d\tau_{1}d\tau_{2}$$
(5)$$E_{Excited}=E_{1}+E_{2}+\iint \;\varphi_{1}{}^{*}\varphi_{2}\left(V_{12}\right)\varphi_{1}{}^{*}\varphi_{2}d\tau_{1}d\tau_{2}\pm \iint \;\varphi_{1}{}^{*}\varphi_{2}\left(V_{12}\right)\varphi_{1}\varphi_{2}{}^{*}d\tau_{1}d\tau_{2}$$

As for charge transportation, the relative organization within the aggregate material can have significant effects on the carrier mobility including influencing the formation of trap states either through deformations, affectations to the packing structure, or through the HOMO and LUMO energy levels [201]. The effective changes to these frontier orbital energies affect the energy cascades and either enhance or hinder charge separation, carrier generation, and subsequent transport properties [202,203]. In addition, the sensitivity of the charge transfer characteristics in relation to geometry has a relevant effect on the excitonic coupling between neighboring molecules and the charge transfer of the system, as well as the formation of charge separated states [204]. This dissociation of electron from hole, and subsequent carrier generation is affected by the formation of J, H, and hybrid aggregates (JH etc.), each having its own unique properties. In lieu of this information, the importance of accurate geometries with which we can examine these non-bonded vdW dominant systems is all the more apparent. 

From the readily apparent importance of having vdW sensitive calculation methods, it is no surprise to find that the development and refinement of accurate dispersion-based calculation methods have been an ongoing process, especially since the turn of the century. Particularly with the use of DFT, which we will focus on here as its effectiveness and scalability makes it a good candidate for aggregation calculations, there are a multitude of options from which to choose.

The development of dispersion specific functionals stems from the inability of older DFT methodologies to capture the behavior of dispersion dominant systems. These DFT methods fail to characterize non-covalent interactions due to their approximation of the exchange-correlation (XC) functional [205,206,207]. The resultant exchange correlation energy of Kohn Sham DFT can be calculated either using a local density approximation (LDA) in which the XC energy is calculated using the density of a uniform electron gas, generalized gradient approximations (GGAs) in which the gradient of the electron density is taken into account in addition to the density, meta-GGAs that additionally utilize the Kohn-Sham kinetic energy, hybrid functionals which utilize a fraction of the Hartree Fock exact-exchange using Kohn Sham orbitals, or range separated functionals in which electron-electron interactions are distance partitioned to prevent self-interaction between electrons [208,209,210,211,212,213]. In systems where—stacking and by extension dispersion is the dominant stabilizing force, GGAs, meta-GGAs, and hybrids all show the force to be solely repulsive, whereas LDAs are relatively accurate. LDAs, however, have been shown to be overly attractive in other purely dispersion-based systems, such as inert gas dimers [214,215,216].

The devised solutions to the miscalculation of the XC functional that we will address include: Grimme’s DFT-D3 correction (an ad-hoc correction), highly parameterized functionals trained upon large sets of data (Minnesota functionals), APFD, and XDM. The XDM model, standing for Exchange-Hole Dipole Moment, is based around the idea that dispersion energies arise from the interaction between electrons and their corresponding ‘exchange holes’. XDM has shown to be relatively accurate in predicting the binding energy of the dimers in the S12L and L7 data sets as well as properly identifying the majority of crystal structures with the lowest energy [217]. Grimme’s D3 dispersion correction was developed in succession to D2 and the original D functional, and has a similar structure involving a scaling factor, a damping function and a heteroatomic coefficient based on the interaction between two atoms. The differences between D3 and D2 lie in an additional C8 term, a slightly different dampening function based on those by Chai and Head-Gordon, as well as the lack of a scaling factor for the first term. The D3 correction has had varying results, some of which say its improvement is non-differentiable from D2 with functionals such as PBE or B3LYP, and others stating that these functionals are not among the best used for thermochemical calculations using Grimme’s D3 [218,219,220]. The Minnesota Functionals, which are highly parameterized and trained on large subsets of data, have a variety of recommendations on what is considered ‘state of the art’, based on what data set is being tested and for what value is being tested for. M06-HF for instance being recommended for the electronic structure community and for time-dependent density functional theory (TDDFT) calculations as it was designed to be accurate for charge transfer excitation energies [221,222,223]. APFD is based off a spherical atom model which is applied to the APF functional which is designed to be dispersionless, the result of which is a seemingly larger dispersion correction than either Grimme’s D3 or M06-2X. APFD itself is comparable to CCSD(T)/aug-cc-pvTZ for noble gas dimers and is comparable to other dispersion corrections concerning hydrogen bonding complexes. It has been noted that the computational cost of APFD in comparison to some D3 methods and CCSD(T) is reduced significantly. However, certain small hydrocarbons such as benzene can have their binding energies overestimated [209]. While the relative energy changes for different systems vary from none impact to significant for the purpose of rational design of new materials, good DFT methods that can describe accurately the vdW interactions are needed [113,224].

Interestingly, or perhaps expectedly as many of the dispersion corrected methods were not necessarily designed with this as their first intention, utilizing these functionals to estimate the frontier orbital energies of aggregates is a mixed bag. For small organic hydrocarbons M06-2X is wildly inaccurate, Grimme’s D3 sees no change in the HOMO-LUMO gap (H-L Gap) although the HOMO and LUMO values both differ (both increase), and APFD does see a decrease in the HOMO-LUMO gap from its calculation of monomer systems, but the calculated H-L gap values (at least for small organics) seems to be similar to that of B3LYP-D3 or B3LYP. The frontier orbitals for these small hydrocarbons do show non-degenerate values corresponding to the HOMO and LUMO of each molecule, but all the values show these orbitals to be of higher energy. This lends credence to the dubious nature of the accuracy of these calculations. Perhaps M06-HF would fare better in this regard. Accepting this, it becomes readily apparent that the current dispersion-based methods are not adequately equipped to help accurately predict many of the spectral curiosities seen through aggregation, although they still may be useful in identifying conformational behavior in regard to aggregative organic materials. Current methods, utilizing DFT to examine the single molecule structure and not the aggregative bulk, still provide enough supplemental insight and are computationally cheap enough to be considered the ‘best’ option.

In summary, the DFT methods for describing the properties of organic aggregates are still at the assessment stage and witnessing growing efforts towards this direction it becomes apparent that the importance of having accurately described vdW interactions is tantamount. Optimizing our methodologies and modes of modeling will not only serve to enlighten our understanding of aggregates in regard to optoelectronic devices, but also other systems including biological systems, chemistry of condensed phases, and surface chemistry; vdW interactions are particularly important in examining systems such as aggregates where weak interactions are relevant.

## 5. Conclusions

The purpose of this review was to summarize the current progress of dye aggregation in DSSCs. The interaction between dye molecules can lead to the formation of H-aggregates and J-aggregates, which are associated with hypsochromic and bathochromic of the light absorption spectra, respectively. The disadvantage of aggregation in quenching the excited state of dye molecules makes researchers focus on inhibiting aggregation, including molecular engineering, the use of co-adsorption agents or adjust of sensitization conditions. However, it is worth noting that the broadening of absorption spectrum, which is one of the focuses of many DSSCs studies, is the inherent advantage of dye aggregation, especially J aggregation.

In the current study, only a very small amount of dye aggregation can improve the performance of DSSCs, which is due to the competitive relationship between the advantages and disadvantages of dye aggregation. So far, it is different to formulate general molecular design rules to categorize good or bad attributes of dye aggregation for DSSCs. Also, because there are many factors that affect dye aggregation, there is no universal law to predict the pattern of dye aggregation. Theoretical calculation will be a powerful tool to predict the pattern of dye aggregation on the TiO_2_ surface. In conclusion, it is a potential approach to improve device performance by using dye aggregation, and the combination of theoretical calculation and aggregation regulation is helpful to achieve this goal.

## Figures and Tables

**Figure 1 molecules-25-04478-f001:**
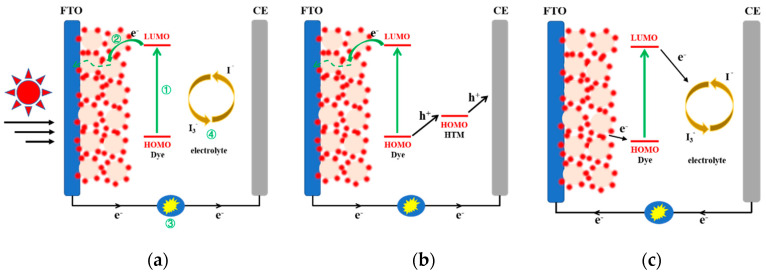
Schematic diagram of charge transport mechanisms in (**a**) liquid n-type DSSCs; (**b**) solid-state n-type DSSCs); (**c**) liquid p-type DSSCs.

**Figure 2 molecules-25-04478-f002:**
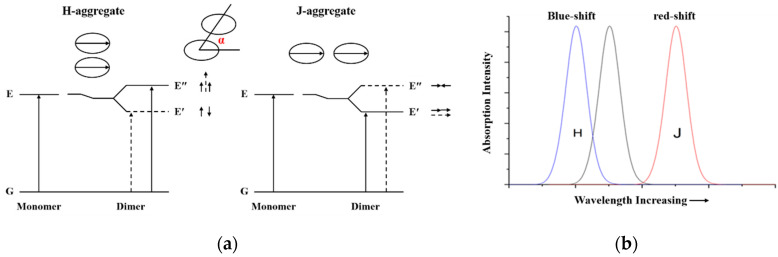
(**a**) Schematic of exciton theory for a molecular dimer to explain the changes in the absorption spectrum; (**b**) The relative shift of the absorption intensity for H and J aggregates in relation to their monomeric form.

**Figure 3 molecules-25-04478-f003:**
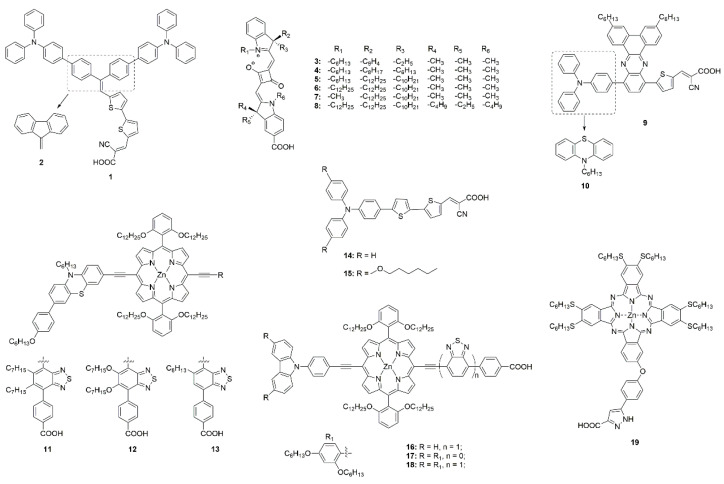
Chemical structures of dyes **1**–**19**.

**Figure 4 molecules-25-04478-f004:**
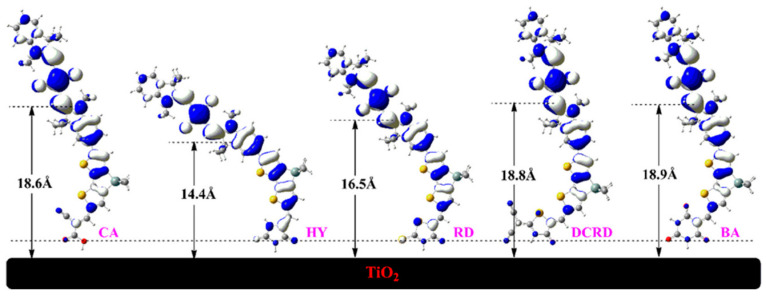
Illustration of molecular space stretching and distance from the electron recapture center to TiO_2_ surface. Reprinted from [115], with permission from Elsevier.

**Figure 5 molecules-25-04478-f005:**
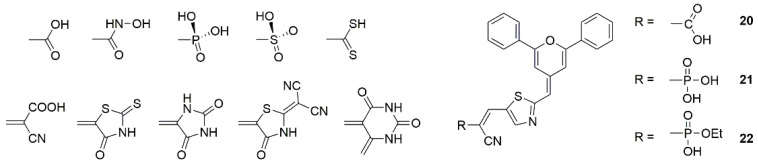
Chemical structures of anchoring groups and dyes **20**–**22**.

**Figure 6 molecules-25-04478-f006:**
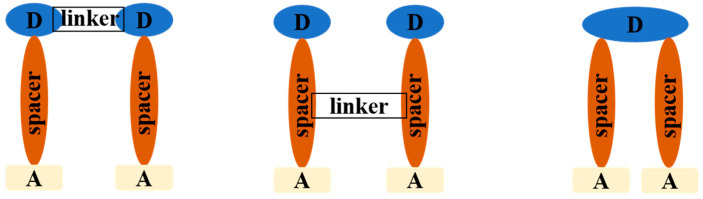
Schematic diagram of double branched dyes.

**Figure 7 molecules-25-04478-f007:**
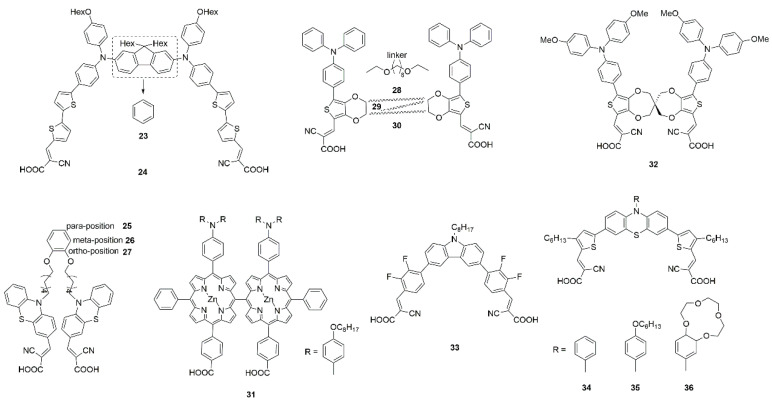
Chemical structures of dyes **23**–**36**.

**Figure 8 molecules-25-04478-f008:**
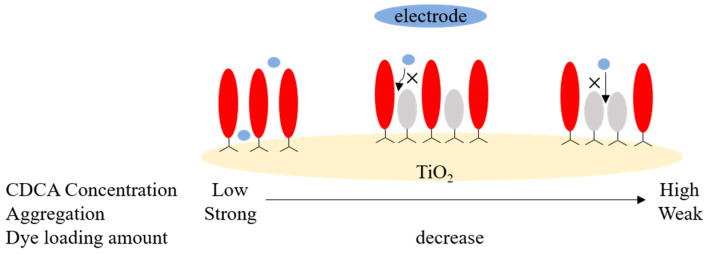
Diagram of the effect of co-adsorbent on dye aggregation.

**Figure 9 molecules-25-04478-f009:**
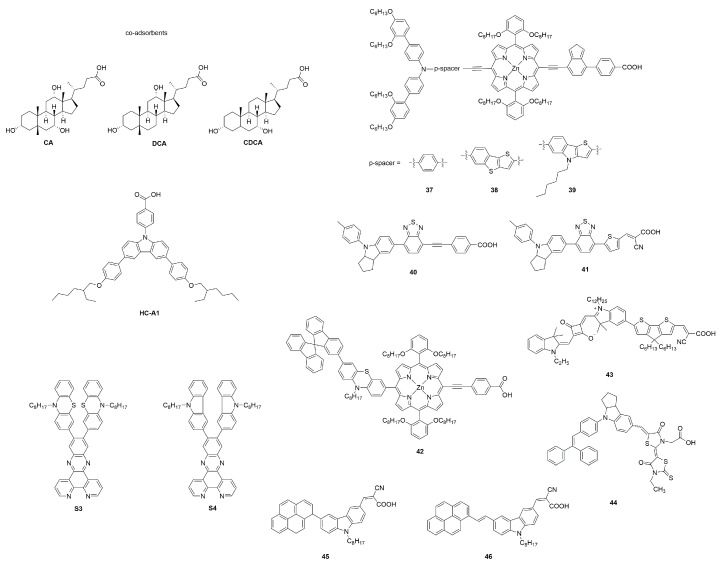
Chemical structures of co-adsorbents and dyes **37**–**46**.

**Figure 10 molecules-25-04478-f010:**
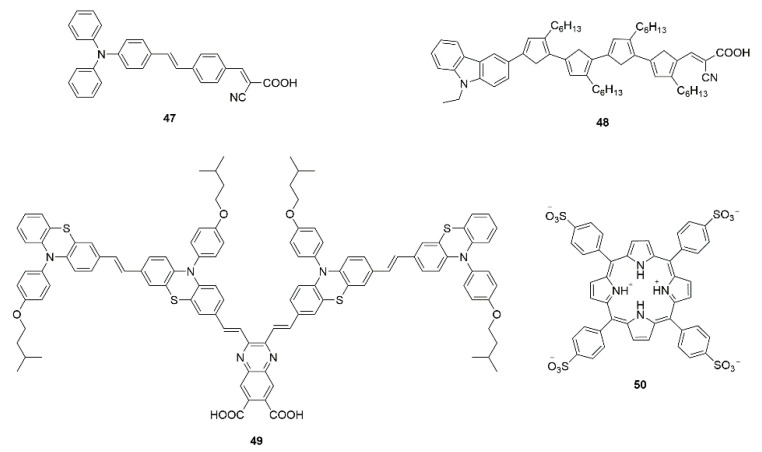
Chemical structures of dyes **47**–**50**.

**Figure 11 molecules-25-04478-f011:**
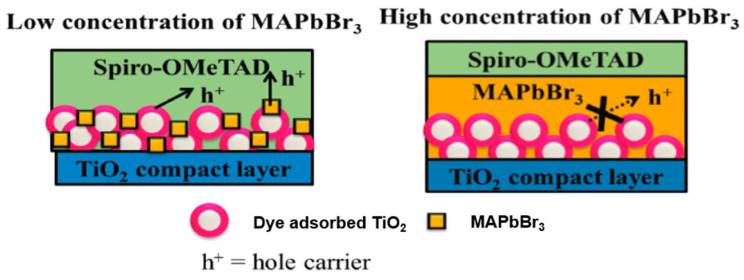
Schematic representation of proposed electron transport mechanism [104].

**Figure 12 molecules-25-04478-f012:**
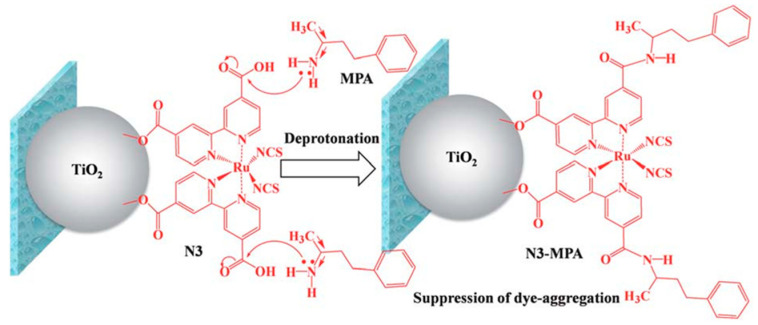
Schematic diagram for deprotonation of dye N3 (i.e., dye **51**) on TiO_2_ particles with MPA via an amide bond. Republished with permission of Royal Society of Chemistry from [95], permission conveyed through Copyright Clearance Center, Inc.

**Figure 13 molecules-25-04478-f013:**
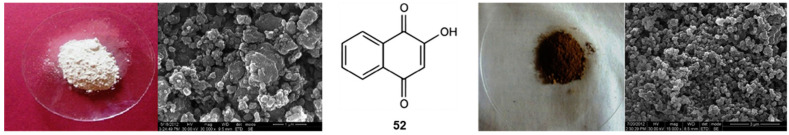
Nanoparticle powder and SEM images of pure TiO_2_ and pre dye **52** treated TiO_2_. Reprinted from [96], with permission from Elsevier.

**Figure 14 molecules-25-04478-f014:**
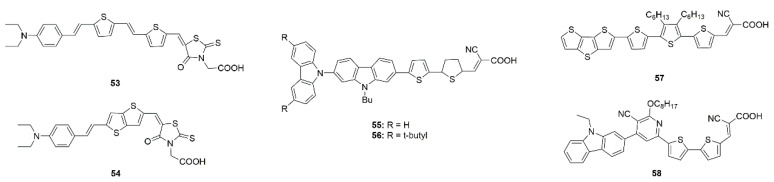
Chemical structures of dyes **53**–**58**.

**Figure 15 molecules-25-04478-f015:**
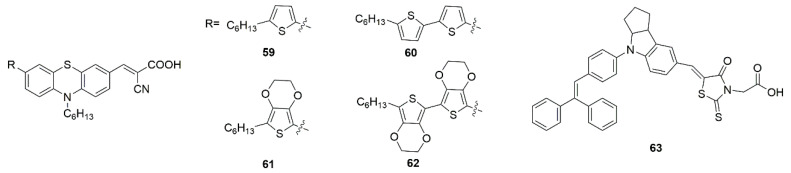
Chemical structures of dyes **59**–**63**.

**Table 1 molecules-25-04478-t001:** The fabrication conditions and photovoltaic performance of DSSCs and ssDSSCs.

DSSCs									
Dye	Dye-Bath Solvent ^1^ (Co-Adsorbents)	λ_max_/nm ^2^	λ_max_/nm ^3^	Dye Loading Amount (10^−7^ mol cm^−2^)	J_sc_/mA cm^−2^	V_oc_/V	FF	η/%	Ref.
**1**	CHCl_3_/MeOH	307, 344, 485	441	-	10.36	0.715	0.722	5.35	[71]
	CHCl_3_/MeOH (DCA)	-	441	-	10.02	0.714	0.712	5.09	
**2**	CHCl_3_/MeOH	306, 368, 468	454	-	6.87	0.687	0.678	3.20	
	CHCl_3_/MeOH (DCA)	-	450	-	7.48	0.683	0.734	3.75	
**3**	ACN/CHCl_3_	643	-	2.16	7.64	0.668	0.710	3.62	[73]
**4**	ACN/CHCl_3_	643	-	2.29	8.89	0.683	0.770	4.67	
**5**	ACN/CHCl_3_	643	-	2.00	10.95	0.706	0.750	5.80	
**6**	ACN/CHCl_3_	643	-	1.96	11.55	0.715	0.700	5.78	
**7**	ACN/CHCl_3_	642	-	2.47	8.78	0.671	0.770	4.53	
**8**	ACN/CHCl_3_	650	-	1.63	11.95	0.717	0.710	6.08	
**9**	ACN/TBA/DMSO	378, 496	-		14.11	0.660	0.653	6.08	[75]
**10**	ACN/TBA/DMSO	384, 494	-		14.32	0.910	0.539	7.04	
**11**	Toluene/EtOH	465, 620, 675	-	1.55	10.51	0.700	0.719	5.19	[76]
**12**	Toluene/EtOH	465, 621, 678	-	1.62	12.79	0.701	0.716	6.42	
**13**	Toluene/EtOH	465, 623, 683	-	1.67	17.93	0.711	0.715	9.12	
**14**	THF/EtOH	-	-	0.782	8.60	0.604	0.690	3.60	[77]
	THF/EtOH (CDCA)	-	-	0.537	8.90	0.610	0.680	3.70	
	THF/EtOH ^4^	-	-	0.776	10.70	0.650	0.700	4.90	
**16**	CHCl_3_/EtOH	432, 459, 580, 646	634	0.525	11.60	0.760	0.710	6.26	[78]
**17**	CHCl_3_/EtOH	446, 575, 628	626	0.358	11.47	0.860	0.670	6.60	
**18**	CHCl_3_/EtOH	431, 457, 585, 646	640	0.402	12.50	0.781	0.720	7.03	
**19**	EtOH/ACN	697	705	0.307	3.98	0.601	0.700	1.67	[79]
	EtOH/ACN (CDCA)	-	-	0.276	4.27	0.611	0.720	1.89	
**23**	THF/EtOH	458	-	-	2.50	0.520	0.780	1.00	[80]
	THF/EtOH (DCA)	-	-	-	8.90	0.600	0.760	4.05	
**24**	THF/EtOH	494	-	-	9.25	0.630	0.610	3.56	
	THF/EtOH (DCA)	-	-	-	10.8	0.650	0.600	4.20	
**25**	CH_2_Cl_2_/MeOH	305, 424	422	8.12	12.26	0.756	0.660	6.14	[81]
**26**	CH_2_Cl_2_/MeOH	305, 414	412	9.48	11.92	0.740	0.660	5.85	
**27**	CH_2_Cl_2_/MeOH	305, 420	422	10.57	10.92	0.705	0.680	5.25	
**28**	Toluene	501	419	-	14.83	0.755	0.720	8.10	[82]
**29**	Toluene	494	416	-	13.21	0.756	0.750	7.50	
**30**	Toluene	488	414	-	12.00	0.752	0.730	6.60	
**31**	THF	422, 464, 569, 615	-	-	13.20	0.650	0.620	5.33	[83]
**32**	DMSO	437	464	-	14.00	0.570	0.690	5.51	[84]
**33**	N/A	253, 290, 401	453	-	10.20	0.707	0.592	5.20	[85]
**34**	ACN/TBA	477	-	2.88	15.64	0.667	0.670	7.02	[86]
**35**	ACN/TBA	488	-	2.26	18.16	0.680	0.650	7.99	
**36**	ACN/TBA	487	-	2.38	18.19	0.706	0.690	8.82	
**37**	THF/EtOH (CDCA)	-	-	0.238	13.59	0.759	0.772	8.30	[87]
	THF/EtOH (HC-A1)	-	-	0.220	15.62	0.759	0.762	9.05	
**38**	THF/EtOH (CDCA)	-	-	0.208	15.58	0.858	0.738	9.87	
	THF/EtOH (HC-A1)	-	-	0.204	16.42	0.846	0.769	10.69	
**39**	THF/EtOH (CDCA)	-	-	0.199	15.82	0.858	0.731	9.94	
	THF/EtOH (HC-A1)	-	-	0.186	16.50	0.846	0.772	10.80	
**40**	CHCl_3_/EtOH	508	474	2.33	13.44	0.786	0.675	7.13	[88]
	CHCl_3_/EtOH (CDCA)	-	-	2.32	15.06	0.775	0.704	8.21	
	CHCl_3_/EtOH (dye 41)	-	-	1.87 + 0.45	18.30	0.737	0.729	9.83	
**41**	CH_2_Cl_2_	318, 405, 546	-	-	8.46	0.601	0.760	3.86	[89]
	CH_2_Cl_2_ (CDCA)	-	-	-	17.50	0.657	0.740	8.56	
	CHCl_3_	-	-	-	16.25	0.618	0.720	7.22	
	CHCl_3_ (CDCA)	-	-	-	17.82	0.646	0.720	8.29	
N719	EtOH	-	-	-	0.22	0.600	0.230	0.03	[90]
	EtOH/H_2_O	-	-	-	4.15	0.650	0.570	1.50	
**45**	TBA/ACN				8.07	0.800	0.760	4.90	[91]
	TBA/ACN (CDCA)				8.08	0.790	0.680	4.34	
**46**	TBA/ACN				11.57	0.810	0.590	5.57	
	TBA/ACN (CDCA)				11.59	0.800	0.680	6.30	
**47**	CH_2_Cl_2_	438	423	4.46	9.70	0.760	0.720	5.33	[92]
	ACN	-	-	4.39	9.40	0.720	0.680	4.59	
	EtOH	-	-	3.36	9.10	0.709	0.660	4.23	
	THF	-	-	3.25	8.20	0.663	0.670	3.61	
	DMF	-	-	0.69	5.60	0.579	0.620	2.00	
**49**	THF ^5^	-	-	1.11	12.73	0.650	0.680	5.60	[93]
	THF ^6^	-	-	1.25	11.83	0.640	0.700	5.29	
**50**	ACN/H_2_O ^7^	-	-	-	0.48	0.220	0.460	0.05	[94]
	ACN/H_2_O ^8^	-	-	-	2.83	0.290	0.610	0.50	
**51**	N/A	-	-	-	14.37	0.740	0.685	7.23	[95]
	N/A ^9^	-	-	-	14.91	0.780	0.715	8.28	
**52**	N/A	-	-	-	1.60	0.480	0.643	1.00	[96]
	N/A ^10^	-	-	-	2.99	0.500	0.669	1.47	
**53**	THF	513	-	-	10.64	0.520	0.700	3.87	[97]
**54**	THF	488	-	-	15.23	0.560	0.730	6.23	
**55**	ACN/TBA/DMSO	262, 294, 345, 476	-	-	15.78	0.601	0.640	6.04	[98]
**56**	ACN/TBA/DMSO	258, 298, 351, 478	-	-	14.00	0.612	0.640	5.48	
**57**	ACN/TBA	314, 429	-	3.69	12.09	0.620	0.670	5.02	[99]
**58**	ACN	285, 405, 475	-	1.92	6.56	0.540	0.689	2.44	[100]
	ACN (CDCA)	-	-	2.04	5.87	0.560	0.686	2.25	
**ssDSSCs**									
**14**	N/A ^11^	438	-	-	0.65	0.719	0.780	3.89	[101]
**15**	N/A ^11^	458	-	-	0.78	0.740	0.730	4.51	
**20**	ACN/TBA	-	-	0.28	6.40	0.710	0.570	2.60	[102]
**21**	ACN/TBA	-	-	-	6.80	0.790	0.430	2.30	
**22**	ACN/TBA	-	-	0.19	7.10	0.790	0.460	2.60	
**44**	ACN	-	-	-	12.90	0.550	0.483	3.40	[103]
	ACN (CA)	-	-	-	13.45	0.552	0.506	3.60	
**44**	TBA/ACN	-	-	0.066	4.70	0.760	0.727	2.60	[104]
	TBA/ACN (MAPbBr_3_)	-	-	0.052	5.40	0.810	0.704	3.10	
**48**	ACN/TBA	439	-	-	16.14	0.496	0.420	3.33	[105]
	Toluene	484	-	-	13.42	0.453	0.390	2.40	

N/A is not available. ^1^ MeOH (methanol); CHCl_3_ (chloroform); ACN (acetonitrile); TBA (tert-butyl alcohol); CH_2_Cl_2_ (dichloromethane); EtOH (ethanol); THF (tetrahydrofuran); DMF (dimethyl formamide); DMSO (dimethyl sulfoxide). ^2^ The absorption data were measured in solvent. ^3^ Dyes were adsorbed on TiO_2_ films. ^4^ Dye loaded TiO_2_ electrode is coated with Al_2_O_3_ by atomic layer deposition. ^5^ Sensitization time 12 h. ^6^ Sensitization time 24 h. ^7^ Add HClO_4_ to the dye bath and electrolyte. ^8^ Add H_2_SO_4_ to the dye bath and electrolyte. ^9^ The N3-sensitized TiO_2_ films are dipped into MPA solution to form N3-MPA on TiO_2_ by deprotonation. ^10^ The pure TiO_2_ was modified by directly mixing the natural dye during synthesis of TiO_2_ to yield pre dye treated TiO_2_ nanoparticles. ^11^ The overall conversion efficiency of DSSCs is tested under lower light intensity 9.3% sun.

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
