# Peer review of "Cause, Regulation and Utilization of Dye Aggregation in Dye-Sensitized Solar Cells"

_molecules, 2020, doi:10.3390/molecules25194478_

Round 1

Reviewer 1 Report

The review by Zhou and coworkers clearly review the effect of the molecular aggregation on the performances of dye-sensitized solar cells (DSSCs). The latter issue is of paramount interest in the enhancement of photovoltaic efficiency in DSSC. The is well organized and written; each subsection is self-consistent. Language is good throughout the manuscript. Very interesting is the section dealing with the "application of quantum computation in the study of dye aggregation" being a meangiful bridge between theoretic and practic chemistry. 

This review has two main concern, one related to some missing discussion, the second one related to the bibliography.

1.i) the review is mainly focused on the aggregation phenomena onto TiO2 surface. This reviewer is aware than TiO2-based DSSCs are the most exploited and efficient ones, yet, to reach a broader audience (not only related to DSSCs) it could be useful to tackle also other n-type semicondutors (SnO, ZnO...) or p-type ones (NiO, CuO...) being the latter the substrate for p-type DSSCs. 

1.ii) Section 2.2 is out of the scope of the present work, it could be more useful to delete that part and recall some recent review about the topic

1.iii) In section 3.3.1 dealing with the molecular engineering as a powerful strategy to minimize dye aggregation, the authors should put more emphasys on the functionalization of a central core (in symmetric dyes) to force the absorption geometry of the dyes. Some literature references are: a) 10.1016/j.jphotochem.2016.01.018 b) 10.1039/C9TC05014K c)  10.1021/acs.jpcc.6b03965 d) 10.1039/D0NJ03228J among others. Please comment on this topic

1.iv) Among the different strategies to reduce the dye aggregation, the authors did not mention the electrode pretreatment. The latter could be a useful approach to reduce the number anchoring sites and, in turns, the dye aggregation. This was proved to be effective both in n-type (10.1021/la050134w; 10.1109/JPHOTOV.2018.2806307; ) and p-type (10.1002/slct.201702867; 10.1021/acsami.7b01532; 10.1021/acsami.6b01090) DSSCs. 

2) With respect to the bibliography, the paper presents a relatively high number of self-citations: see for example the reference cluster 43-58. The self-citation should be reduced and the cluster should be integrated with references coming from other research groups.

All considering the present could be accepted for publication in molecules after the above reported points will be addressed

Reviewer 2 Report

The manuscript entitled "Cause, Regulation and Utilization of Dye Aggregation in Dye-Sensitized Solar Cells" is a well-written compehensive review considering the topic crucial from the point of view of practical development of novel dye-sensitized solar cells. Aggregation phenomena are one of the most troublesome due to the quenching the excited states, however after a proper investigation they can also be applied in order to enhance the desired emission or two-photon cross section. The manuscript covers the different ways of controlling dye aggregation processes and computational modelling of the aggeration processes. However, as the experimental part of the review is advanced and broad, the theoretical section seems to be a little bit trivial and too general. It also does not provide proper literature references on the aggregation processes, but only general remarks on the computational methodology that can be applied to any kind of intermolecular interactions. Therefore in my opinion, the computational section should be rewritten and supplemented with the current progress in computational study of dye aggregation. Exemplary work that can be cited is as follows: https://doi.org/10.1038/srep35893,https://doi.org/10.1016/j.orgel.2016.12.043, https://doi.org/10.1039/C2SC20861J or https://doi.org/10.1002/ejic.201801118 and references therein. 

Upon the adequate modifications of this part of the manuscript, I do recommend it for publication in Molecules.

Reviewer 3 Report

The review presents the operating mechanism of liquid and solid dye solar cells as well as the mechanism of dye aggregation and the influence of various factors on dye aggregation. This review is clearly structured, interesting and suitable for publication after some changes.

In introduction part a general overview of new developments in DSSCs should be given. For example, some research groups are working on textile-based or nanofiber-based dye solar cells.

Line 143: Fig. 2: In this graphic it is not clear at first glance what the description means. Was " blue-schift" rather meant as "blue-shift of absorption spectrum"? It should be changed.

Line 357: At this point “traditional” dye co-sensitization is mentioned, what exactly is meant? The short explanation should be inserted here.

Line 171: Here various dyes are mentioned. Which dyes are these and how are they based? For example benzothiazole-based dyes? At this point it might be better to explain more clearly.

Line 364: At this point three types of methods are mentioned. Which methods are meant here? It would be better for convenience to briefly mention these three types of methods.

Line 435: In this review a quantum calculation is mentioned for the study on dye aggregation. At this point an extended overview of other computer-based approaches should be presented. Could it be, for example, a multiscale approach, such as Quantum Mechanics/Molecular Mechanics (QM/MM)? QM/MM is normally often used to study biomolecules or DFT+U approach, which can be used to improve the description of the electronic structure of metal oxides or PBS calculations?

Round 2

Reviewer 1 Report

The authors precisely reply to all reviewers' comments and implement their manuscript accordingly. 

The manuscript could now published in "Molecules"